# Sex Determination Using RNA-Sequencing Analyses in Early Prenatal Pig Development

**DOI:** 10.3390/genes10121010

**Published:** 2019-12-05

**Authors:** Susana A. Teixeira, Adriana M. G. Ibelli, Maurício E. Cantão, Haniel C. de Oliveira, Mônica C. Ledur, Jane de O. Peixoto, Daniele B. D. Marques, Karine A. Costa, Luiz. L. Coutinho, Simone E. F. Guimarães

**Affiliations:** 1Animal Science Department, Universidade Federal de Viçosa, Viçosa 36570-900, Brazil; susana.amaral.teixeira@gmail.com (S.A.T.); hanielcedraz@gmail.com (H.C.d.O.); danielebdiniz@gmail.com (D.B.D.M.); kryneacosta@yahoo.com.br (K.A.C.); 2Animal Genetics Laboratory, Embrapa Swine and Poultry National Research Center, Concórdia 89715-899, Brazil; adriana.ibelli@embrapa.br (A.M.G.I.); mauricio.cantao@embrapa.br (M.E.C.); monica.ledur@embrapa.br (M.C.L.); jane.peixoto@embrapa.br (J.d.O.P.); 3Functional Genomics Center, ESALQ, Universidade de São Paulo, Piracicaba 89715-899, Brazil; llcoutinho@usp.br

**Keywords:** early gestation, conceptuses development, conceptuses sex determination, pig transcriptome

## Abstract

Sexual dimorphism is a relevant factor in animal science, since it can affect the gene expression of economically important traits. Eventually, the interest in the prenatal phase in a transcriptome study may not comprise the period of development in which male and female conceptuses are phenotypically divergent. Therefore, it would be interesting if sex differentiation could be performed using transcriptome data, with no need for extra techniques. In this study, the sex of pig conceptuses (embryos at 25 days-old and fetuses at 35 days-old) was determined by reads counts per million (CPM) of Y chromosome-linked genes that were discrepant among samples. Thus, ten genes were used: *DDX3Y*, *KDM5D*, *ZFY*, *EIF2S3Y*, *EIF1AY*, *LOC110255320*, *LOC110257894, LOC396706, LOC100625207,* and *LOC110255257*. Conceptuses that presented reads CPM sum for these genes (ΣCPM_chrY_) greater than 400 were classified as males and those with ΣCPM_chrY_ below 2 were classified as females. It was demonstrated that the sex identification can be performed at early stages of pig development from RNA-sequencing analysis of genes mapped on Y chromosome. Additionally, these results reinforce that sex determination is a mechanism conserved across mammals, highlighting the importance of using pigs as an animal model to study sex determination during human prenatal development.

## 1. Introduction

Maternal nutrition may affect progeny anatomy, physiology and metabolism during critical periods of prenatal life, which is described as fetal programming [1]. The effects of maternal nutrition on fetal programming have been recently addressed in several studies using pigs as a model organism [2,3,4,5,6], since there are striking similarities between pigs and humans regarding the anatomy, physiology, metabolism, and nutrition, which provide the basis for the use of this animal in several studies. In addition, *Sus scrofa* is a prolific specie with a relatively short developmental period, besides being a cheaper and easier mammalian model organism to maintain [7,8]. In this context, transcriptome studies using RNA-sequencing (RNA-seq) performed in pigs highlight the relevance of the data obtained in this specie for human-related researches. 

The RNA sequencing (RNA-seq) approach generates a comprehensive picture of gene expression levels at different developmental stages and physiological conditions [9,10,11]. However, these analyses may be influenced by sexual dimorphism, since males and females are marked by significant biological differences that can modify the gene expression of economically important traits [12]. Therefore, researchers should take into account the conceptuses sex and sex ratios in their research before drawing any conclusion [13].

In mammals, sex determination is basically organized in four stages: (1) Determination of the chromosomal sex, which is established at fertilization when a Y- or a X-chromosome sperm fuses with the oocyte to determine the zygote genetic sex; (2) the differentiation of gonads into testicles or ovaries from the presence or absence of sex-determining region Y gene (*SRY*) at the critical time window during embryonic development; (3) the differentiation of male and female internal and external genitalia from the non-sexually dimorphic structures present in the embryo, and (4) anatomical and physiological differences, i.e. sexual differentiation. Thus, all sex determination stages are mainly related to general gonadal function [14,15]. 

At early pig development, the absence of gonadal phenotype may impair the accurate visual sex determination, since the tunica albuginea, a marker of testis formation, is histologically identified only at 27 days [16] and the beginning of testicular descent occurs around 60 days [17]. Therefore, if research projects are designed at early development stages in which male and female conceptuses are not phenotypically divergent and there is response to sexual dimorphism, the sex determination could be performed using alternative tools rather than phenotypic differences.

Several studies have demonstrated how to identify the conceptuses sex at early stages of development using molecular techniques, as PCR (polymerase chain reaction) from Y chromosome-linked genes [13,18,19,20]. The PCR provides sensitive, precise, rapid, and reliable results [20]. However, if a RNA-seq study has already been performed on conceptuses to obtain a broader knowledge of the transcriptome under experimental conditions (e.g. inclusion of additives on maternal diets), these data can also be applied to determine the conceptuses sex with no need for extra techniques. A similar approach has been used by Petropoulos et al. [21] to determine the sex of human embryos through single-cell sequencing.

Therefore, considering that in early prenatal pig development male and female conceptuses are not phenotypically divergent and the sexual dimorphism may affect the transcriptome analyses, we aimed to demonstrate that the sex of 25 and 35 days-old conceptuses can be determined by read counting of genes mapped on Y chromosome using already available RNA-seq data.

## 2. Materials and Methods 

### 2.1. Experimental Animals and Design

The experimental protocols used in this study have followed ethical principles in animal research (CONCEA, 2016) and were approved by the Ethical Committee on Animal Use of the *Universidade Federal de Viçosa* (UFV), MG, Brazil [protocol # 06/2017]. 

All experimental protocols were performed according to the experimental design described in Costa et al. [22]. Briefly, 24 hours after the second insemination, 11 gilts received a basal diet for pregnant animals without supplementation (CONT) and 12 gilts received a CONT diet supplemented with 1.0% L-arginine (ARG), and two gestational ages were considered (25 and 35 days). From these gilts, 20 became pregnant. Additional details regarding reproductive management and nutritional information of the diets have been previously described in Costa et al. [22].

At 25 days of gestation, five females of CONT (*n* = 5) and five females of ARG (*n* = 5) were rendered unconscious using head-only electrical stunning (240V, 1.3A) and immediately exsanguinated. The same procedure was followed for four females of CONT (*n* = 4) and six females of ARG (*n* = 6) at 35 days of gestation. After slaughter, four conceptuses were collected per female at each gestational age, totaling in average 20 conceptuses per treatment. 

The conceptuses were quickly washed with PBS (Phosphate Buffered Saline) solution, individually identified, stored in liquid nitrogen and transported to the Animal Biotechnology Laboratory (LABTEC) at the Department of Animal Science, UFV. At LABTEC, each conceptus (embryos from gilts slaughtered at 25 days and fetuses from gilts slaughtered at 35 days) was entirely and separately macerated in liquid nitrogen. Although an average of five gilts was initially used per treatment in the original trial described in Costa et al. [22], subsequent analyses were performed on a subset of gilts (*n* = 3) per treatment, since at least three biological replicates are recommended for describing results in RNA-seq experiments [9,23]. Moreover, three embryos from each CONT gilt (*n* = 3 gilts) and three embryos from each ARG gilt (*n* = 3 gilts) at 25 days of gestation (25DC and 25DA, respectively) and three fetuses from each CONT gilt (*n* = 3 gilts) and three fetuses from each ARG gilt (*n* = 3 gilts) at 35 days of gestation (35DC and 35DA, respectively), totaling 36 samples, were randomly chosen for RNA-seq analysis. Afterwards, these samples were transported in liquid nitrogen to the Animal Genetics Laboratory at the Embrapa Swine and Poultry National Research Center, Concordia, SC, Brazil, for further RNA extraction and library preparation.

### 2.2. RNA Extraction and Library Preparation

Total RNA extraction of the 25- and 35-day-old conceptuses from the ARG and CONT females was performed with TRIzol (Invitrogen, San Diego, CA, USA). The macerated conceptuses (100 mg) and TRIzol (1 mL) were mixed with vortex and then incubated for 5 minutes at room temperature (RT, 25 °C). Then, 200 µL of chloroform were added, shaking vigorously for 15 seconds and incubated at RT for 5 minutes. Centrifugation was performed at 11,000× g at 4 °C for 15 minutes. Approximately 600 μL of the clear upper aqueous phase containing only RNA were carefully removed and transferred to a new tube, and 600 μL of 70% ethanol were added and homogenized by inversion. This volume was added to the silica column RNeasy mini kit (Qiagen, Hilden, Germany) and centrifuged for 15 seconds at 8000× *g*. The eluate was discarded and 700 μL RW1 buffer were added, followed by centrifugation for 15 seconds at 8000× g. Two washes with 500 μL RPE buffer were done and, finally, RNAs were eluted in 50 μL of RNase free water.

After RNA extraction, the quantification was performed in a QUBIT fluorimeter (Thermo Scientific, Waltham, MA, USA) and the integrity was determined in 1.0% agarose gel. In addition, the Agilent 2100 BioAnalyzer (Agilent Technologies, Santa Clara, CA, USA) was used for integrity measurement, in which samples with RNA integrity number (RIN) higher than eight were used for library preparation. In this sense, one fetus sample was excluded from subsequent analysis due to low RIN score (RIN < 8). Therefore, the 35 remaining conceptuses samples (nine ARG and nine CONT at 25 days and eight ARG and nine CONT at 35 days) were submitted to RNA-seq library preparation using the TruSeq Stranded mRNA Library Prep Kit (Illumina, Inc., San Diego, CA, USA), followed by purification of the poly-A tail using 2 μg of total RNA, according to the manufacturer´s recommendations.

### 2.3. Sequencing, Quality Control and Mapping

The libraries were sequenced in Illumina HiSeq2500 (Illumina, Inc.; San Diego, CA, USA), following the 2 × 100 bp paired-end protocol, at the Functional Genomics Center, ESALQ, *Universidade de São Paulo*, Piracicaba, SP, Brazil. The FASTQ files were deposited in the SRA database, with Bioproject number PRJNA576701 and Biosample numbers SAMN13003023, SAMN13003024, SAMN13003025, SAMN13003026, SAMN13003027, SAMN13003028, SAMN13003029, SAMN13003030, SAMN13003031, SAMN13003032, SAMN13003033, SAMN13003034, SAMN13003035, SAMN13003036, SAMN13003037, SAMN13003038, SAMN13003039, SAMN13003040, SAMN13003041, SAMN13003042, SAMN13003043, SAMN13003044, SAMN13003045, SAMN13003046, SAMN13003047, SAMN13003048, SAMN13003049, SAMN13003050, SAMN13003051, SAMN13003052, SAMN13003053, SAMN13003054, SAMN13003055, SAMN13003056 and SAMN13003057.

The BAQCOM pipeline () [24] was used to perform quality control (QC), mapping and reads counting. BAQCOM uses a set of software to analyze RNA-seq data, as Trimmomatic [25], version 0.38, to identify and remove adapter and low quality sequences, HISAT2 [26], version 0.11.2, to map reads against the genome and HTseq-count [27], version 0.11.2, to count reads in features. Using BAQCOM pipeline, only reads with Phred quality ≥ 20 and length ≥ 70 pb were mapped against the pig reference genome (*Sus scrofa*, v. 11.1) and the reads counting for each feature was based on Ensembl annotation release 95.

### 2.4. Sex Identification

The conceptuses sex determination (*n* = 18 embryos and *n* = 17 fetuses) was performed using discrepant reads counts per million (CPM) of Y chromosome-linked genes from the conceptuses RNA-seq dataset. After the samples classification as males or females, the HTseq files containing all samples transcripts were renamed and used to build a multi-dimensional scale (MDS) plot using plotMDS function of *EdgeR* package [28] of R software [29] to evaluate conceptuses dispersion.

## 3. Results

Averages of 15.3 and 14.4 million reads/samples were generated for embryos and fetuses, respectively. After the data QC, averages of 13.6 million reads/embryos and 12.9 million reads/fetuses remained for further analyses. More than 98% of the reads were mapped against the pig reference genome (*Sus scrofa*, v.11.1), with 83.8% and 79.8% of the reads counted into genes for embryos and fetuses, respectively. An average of 0.05% of the reads was uniquely mapped on Y chromosome. At 25 and 35 days of age, male and female conceptuses could not be phenotypically differentiated, since gonadal phenotype could not be observed (Appendix A). 

From 79 Y chromosome-linked genes, ten showed discrepant reads CPM among samples, i.e, high and low reads CPM values (Table 1 and Table 2), and were selected for sex determination: *DDX3Y* (ATP-dependent RNA helicase DDX3X), *KDM5D* (Lysine demethylase 5D), *ZFY* (Zinc Finger Protein, Y-Linked), *EIF2S3Y* (eukaryotic translation initiation factor 2 subunit 3), *EIF1AY* (eukaryotic translation initiation factor 1A, Y-linked), *LOC110255320* (lysine-specific demethylase 6A-like), *LOC110257894* (gamma-taxilin-like), *LOC396706* (U2 small nuclear ribonucleoprotein auxiliary factor 35 kDa subunit-related protein 2), *LOC100625207* (probable ubiquitin carboxyl-terminal hydrolase FAF-X) and *LOC110255257* (oral-facial-digital syndrome 1 protein-like). The CPM sum for these genes (ΣCPM_chrY_) was calculated for each conceptus in order to determine its sex. In this way, conceptuses that presented ΣCPM_chrY_ greater than 400 were classified as males and those with ΣCPM_chrY_ below 2 were classified as females. Male conceptuses showed mean (standard deviation), minimum and maximum ΣCPM_chrY_ of 499.08 (44.74), 427.29, and 573.09, respectively (Table 1), while females showed mean (standard deviation), minimum and maximum ΣCPM_chrY_ of 0.38 (0.42), 0.00 and 1.75, respectively (Table 2). In the MDS plot, a segregation of conceptuses into two distinct sex groups was observed (Figure 1). 

## 4. Discussion

The control of animals sex ratio is desirable in livestock production [20], since the expression of genes that affect a wide range of economically important traits may be influenced by sexual dimorphism [12]. Considering that the sex determination is equivalent to testis determination [30], the testis could be easily used to determine male and female individual rates. However, in pig conceptuses, the tunica albuginea, a marker of testis formation, is histologically identified only at 27 days [16] and the beginning of testicular descent occurs around 60 days [17]. In this context, we demonstrated that it is possible to determine the sex of pig embryos (25 days) and fetuses (35 days) using reads CPM of ten Y chromosome-linked genes through RNA-seq analysis, with no need to use other molecular tools for this purpose, such as PCR. Although most often considered a cheaper and faster technique than RNA-seq, PCR would represent another cost source in this study, since RNA-seq analyses had been previously performed aiming to test a hypothesis on a maternal nutrition experiment [22]. 

Several studies have also used Y chromosome-linked genes to determine the sex of pigs at early development stages [13,18,19], however, all of them have applied DNA-based approaches. The Y chromosome harbors genes that are essential for testis development and function, such as the master gene for testis determination (*SRY)*, which is expressed during a critical period of embryonic development, and other genes that are important for spermatogenic function, as the *ZFY*, *EIF1AY* and *KDM5D*. The importance of these Y-linked genes during male development has been elucidated for mammals [31,32,33,34]. These genes are located in the non-recombining region of the Y chromosome (NRY), which is also known as the male-specific Y (MSY) region, transmitted from fathers to sons without recombination with the X chromosome [35]. The MSY region contains only 27 genes coding for distinct proteins, as the *SRY*, *EIF1AY*, *KDM5D*, *ZFY*, *EIF2S3Y,* and *DDX3Y* [31,35,36]. 

In this study, among all abovementioned genes, *SRY* was not used to differentiate male and female conceptuses (data not shown). This gene seems to follow a strict expression pattern during mammalian embryonic development. In mouse, its expression starts at 10.5 days post coitum (dpc), reaches a peak at 11.5 dpc and then wanes by 12.5 dpc [32,37]. In pigs, the *SRY* expression is detectable in the embryos genital ridge around 21 and 23 days, with faint expression at 26 days and absence of expression at 31 days [38]. Consequently, both 25 and 35 days of pig prenatal periods do not comprise the main *SRY* expression phase, reinforcing the critical time window during embryonic development in which *SRY* is expressed. 

Subsequently to the sex determination by the *SRY* gene, other male-specific genes are important to maintain testicular development and spermatogenesis [39]. According to Bellott et al. [40], during spermatogenesis, each stage of the molecular central dogma is regulated by the following Y-linked genes: *KDM5D*, *ZFY,* and *EIF1AY*. The *KDM5D* gene encodes a histone demethylase enzyme that acts removing tri- and di-methylations of lysine 4 of histone H3 at the start site of transcription in actively transcribed genes [41]. The transcription factor *ZFY*, the first coding gene identified in the human Y chromosome, regulates the transcription of a number of Y-linked genes [31] and mediates multiple aspects of spermatogenesis and reproduction, such as morphology, motility, capacitation, acrosome reaction, and oocyte activation, as well as chromosomal aberrations [42]. The translation initiation factor *EIF1AY* is requested for a high rate of protein biosynthesis and the absence of this gene may contribute to failures in the spermatogenic process [43]. 

Another important translation initiation factor during spermatogenesis is the *EIF2S3Y* gene, located in the short arm of the Y chromosome. This gene has been identified as a key mouse-specific regulator of spermatogonia proliferation and differentiation [44]. Finally, the *DDX3Y* gene, along with *KDM5D* and *EIF1AY*, are azoospermia factors (AZFa) [45]. Azoospermia is defined as the absence of sperm in at least two different ejaculates [31]. The *DDX3Y* is the major AZFa gene and belongs to a highly conserved subfamily of the DEAD-box RNA helicase family [33]. This gene family is involved in oogenesis and spermatogenesis, silencing mobile elements and other repetitive genomic regions in germinal tissues, and is important for gonads formation during embryo development [46]. Therefore, the *KDM5D*, *ZFY*, *EIF1AY*, *EIF2S3Y,* and *DDX3Y* genes showed discrepant reads CPM among samples and have important spermatogenic and gonadal functions, being crucial to differentiate male and female pig conceptuses in the present study. Similar pattern has already been observed in humans blastocyst [21], which highlights the importance of using pigs as an animal model to study sex determination throughout human prenatal development.

The reads CPM observed for the Y-linked genes in female conceptuses may be considered technical artifacts from RNA-seq, since mammalian X and Y chromosomes share regions of high sequence similarity, causing mismapping of short reads to a reference genome [47]. In this context, Ballouz et al. [48] found reads of female samples mapped in Y chromosome, which was also characterized as reads mismapping. Although considered technical artifacts, these Y chromosome reads CPM found in females were not a major issue in the present study, since the discrepant reads CPM among samples for the ten Y chromosome-linked genes were sufficient to differentiate males and females.

The *LOC110255257*, *LOC100625207*, *LOC110255320*, *LOC110257894,* and *LOC396706* genes are located in the Y chromosome, however, they have not yet been fully characterized in *Sus scrofa* genome. According to Skinner et al. [49], there are few data available on the porcine Y chromosome. Therefore, although these genes are located on this chromosome and have shown contribution to sex determination of pig conceptuses due to the discrepant reads CPM among samples, their functions will not be addressed in this study. Further studies might be performed to characterize these Y-linked genes in pigs.

## 5. Conclusions

We have demonstrated that sex identification can be performed from read counts per million of genes mapped in the Y chromosome at early stages of pig development using already available conceptuses RNA-seq data, with no need for extra techniques. Additionally, the results reinforce that molecular sex determination is a mechanism conserved across mammalian species, since the role of these genes is also important during male development in humans, highlighting the applicability of pigs as an animal model to study sex determination during human prenatal development.

## Figures and Tables

**Figure 1 genes-10-01010-f001:**
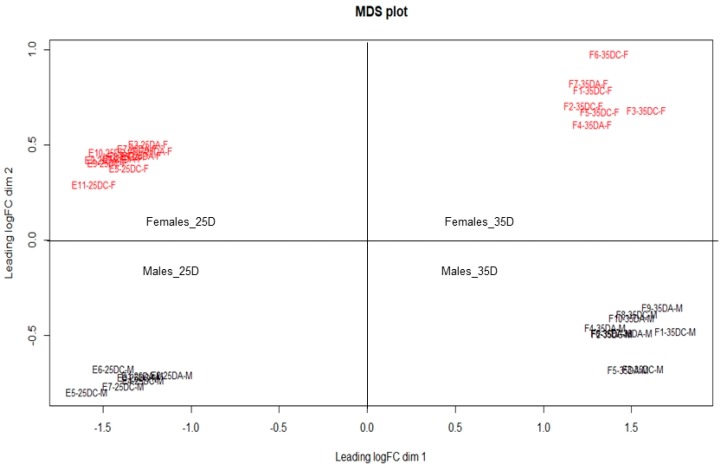
Multi-dimensional scale (MDS) plot showing segregation of pig conceptuses (25D: embryos at 25 days-old and 35D: fetuses at 35 days-old) into two distinct sex groups. Red: Female conceptuses; Black: Male conceptuses; Samples identification: male embryos from supplemented (E-25DA-M) or non-supplemented (E-25DC-M) gilts; male fetuses from supplemented (F-35DA-M) or non-supplemented (F-35DC-M) gilts; female embryos from supplemented (E-25DA-F) or non-supplemented (E-25DC-F) gilts; female fetuses from supplemented (F-35DA-F) or non-supplemented (F-35DC-F) gilts.

**Table 1 genes-10-01010-t001:** Reads counts per million of 10 genes located in the Y chromosome of male pig conceptuses.

Conceptuses Sample_ID ^1^	Counts Per Million (CPM)										ΣCPM_chrY_ ^3^
	*DDX3Y ^2^*	*KDM5D*	*ZFY*	*EIF2S3Y*	*EIF1AY*	*LOC110255320*	*LOC110257894*	*LOC396706*	*LOC100625207*	*LOC110255257*	
E1-25DA-M	151.46	7.52	24.81	162.23	40.64	12.04	9.71	1.53	61.73	15.17	486.84
E2-25DA-M	166.54	7.64	31.68	197.73	45.16	12.30	11.81	1.95	61.00	17.72	553.53
E3-25DC-M	167.21	9.56	28.67	211.97	45.86	13.23	12.79	2.21	66.52	15.07	573.09
E4-25DC-M	170.76	10.15	29.78	189.79	46.39	11.43	12.41	2.11	71.81	17.90	562.53
E5-25DC-M	149.28	6.56	24.94	180.78	36.55	13.47	10.47	1.43	55.29	16.46	495.23
E6-25DC-M	156.64	9.19	30.92	192.23	40.76	12.14	10.99	2.30	68.56	16.24	539.97
E7-25DC-M	166.10	9.43	33.57	192.23	44.92	12.31	11.43	1.84	71.78	17.03	560.64
F1-35DC-M	193.41	5.35	26.07	130.67	44.65	9.79	6.25	1.23	52.46	13.16	483.04
F2-35DC-M	167.28	7.11	28.43	135.38	39.84	10.25	7.19	1.41	56.20	15.04	468.13
F3-35DC-M	168.09	7.05	31.32	141.53	37.80	10.96	7.26	1.78	58.09	13.03	476.91
F4-35DA-M	157.51	8.17	26.77	119.45	33.45	9.30	6.46	2.06	51.27	12.85	427.29
F5-35DA-M	200.33	8.04	24.02	126.13	45.09	12.20	6.56	2.13	45.83	12.20	482.53
F6-35DA-M	187.01	6.47	24.02	128.61	41.58	10.35	8.01	2.02	45.54	13.43	467.04
F7-35DA-M	180.78	7.06	29.27	133.42	41.14	11.74	6.57	2.10	50.78	13.20	476.06
F8-35DC-M	178.15	10.68	33.04	155.57	43.49	11.29	9.23	2.06	62.94	17.32	523.77
F9-35DA-M	143.33	10.28	27.83	135.33	31.42	12.33	8.65	1.55	53.79	12.73	437.24
F10-35DA-M	173.51	10.40	26.69	128.32	36.71	12.62	6.88	1.91	58.19	15.37	470.60

^1^ Conceptuses Sample Identification: Seven male embryos samples from supplemented (E-25DA-M) and non-supplemented (E-25DC-M) gilts; ten male fetuses samples from supplemented (F-35DA-M) and non-supplemented (F-35DC-M) gilts; ^2^
*DDX3Y*: ATP-dependent RNA helicase DDX3X; *KDM5D*: Lysine demethylase 5D; *ZFY*: Zinc Finger Protein Y-Linked; *EIF2S3Y*: eukaryotic translation initiation factor 2 subunit 3; *EIF1AY*: eukaryotic translation initiation factor 1A, Y-linked; *LOC110255320*: Lysine-specific demethylase 6A-like; *LOC110257894*: Gamma-taxilin-like; *LOC396706*: U2 small nuclear ribonucleoprotein auxiliary factor 35 kDa subunit-related protein 2; *LOC100625207*: Probable ubiquitin carboxyl-terminal hydrolase FAF-X; *LOC110255257*: Oral-facial-digital syndrome 1 protein-like; ^3^ ΣCPM_chrY_: sum of reads CPM of 10 Y chromosome-linked genes in male conceptuses.

**Table 2 genes-10-01010-t002:** Reads counts per million of 10 Y chromosome-linked genes that showed reads counting in female pig conceptuses.

Conceptuses Sample_ID ^1^	Counts Per Million (CPM)										ΣCPM_chrY_ ^3^
	*DDX3Y* ^2^	*KDM5D*	*ZFY*	*EIF2S3Y*	*EIF1AY*	*LOC110255320*	*LOC110257894*	*LOC396706*	*LOC100625207*	*LOC110255257*	
E1-25DA-F	0.00	0.07	0.00	0.00	0.07	0.00	0.00	0.00	0.00	0.00	0.14
E2-25DA-F	0.00	0.00	0.00	0.00	0.07	0.00	0.00	0.00	0.00	0.07	0.14
E3-25DC-F	0.00	0.00	0.07	0.07	0.00	0.00	0.00	0.00	0.00	0.00	0.14
E4-25DC-F	0.28	0.00	0.07	0.07	0.07	0.00	0.00	0.00	0.00	0.00	0.49
E5-25DC-F	0.00	0.00	0.00	0.15	0.00	0.00	0.00	0.00	0.07	0.15	0.37
E6-25DC-F	0.00	0.00	0.14	0.00	0.07	0.00	0.00	0.00	0.00	0.00	0.21
E7-25DC-F	0.00	0.00	0.15	0.08	0.00	0.00	0.00	0.00	0.00	0.00	0.23
E8-25DC-F	0.00	0.00	0.00	0.00	0.00	0.00	0.00	0.00	0.00	0.00	0.00
E9-25DC-F	0.00	0.00	0.00	0.00	0.07	0.00	0.00	0.00	0.00	0.07	0.14
E10-25DC-F	0.00	0.00	0.08	0.08	0.08	0.00	0.00	0.00	0.00	0.00	0.24
E11-25DA-F	0.13	0.00	0.07	0.13	0.07	0.00	0.00	0.00	0.20	0.00	0.60
F1-35DA-F	0.08	0.00	0.08	0.00	0.00	0.00	0.00	0.00	0.00	0.08	0.24
F2-35DA-F	0.00	0.08	0.23	0.00	0.08	0.00	0.00	0.00	0.00	0.00	0.39
F3-35DA-F	0.16	0.00	0.00	0.25	0.08	0.00	0.00	0.00	0.08	0.00	0.57
F4-35DC-F	0.00	0.00	0.07	0.00	0.00	0.00	0.00	0.00	0.00	0.00	0.07
F5-35DA-F	0.71	0.00	0.32	0.48	0.08	0.00	0.00	0.00	0.16	0.00	1.75
F6-35DA-F	0.22	0.07	0.14	0.29	0.07	0.00	0.00	0.00	0.22	0.00	1.01
F7-35DA-F	0.00	0.00	0.07	0.00	0.00	0.00	0.00	0.00	0.00	0.00	0.07

^1^ Conceptuses Sample Identification: Eleven female embryos samples from supplemented (E-25DA-F) and non-supplemented (E-25DC-F) gilts; seven female fetuses samples from supplemented (F-35DA-F) and non-supplemented (F-35DC-F) gilts; ^2^
*DDX3Y*: ATP-dependent RNA helicase DDX3X; *KDM5D*: Lysine-specific Demethylase 5D; *ZFY*: Zinc Finger Protein Y-Linked; *EIF2S3Y*: eukaryotic translation initiation factor 2 subunit 3; *EIF1AY*: eukaryotic translation initiation factor 1A, Y-linked; *LOC110255320*: Lysine-specific demethylase 6A-like; *LOC110257894*: Gamma-taxilin-like; *LOC396706*: U2 small nuclear ribonucleoprotein auxiliary factor 35 kDa subunit-related protein 2; *LOC100625207*: Probable ubiquitin carboxyl-terminal hydrolase FAF-X; *LOC110255257*: Oral-facial-digital syndrome 1 protein-like; ^3^ ΣCPM_chrY_: sum of reads CPM of 10 Y chromosome-linked genes in females conceptuses.

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
