# Peer review of "Sex Determination Using RNA-Sequencing Analyses in Early Prenatal Pig Development"

_genes, 2019, doi:10.3390/genes10121010_

Round 1

Reviewer 1 Report

Guimarães et al. use transcriptome data to perform sex determination in the prenatal phase of pigs before conceptuses can be phenotypically differentiated for sex. The study is interesting and can have applications in animal breeding programs. However, it lacks on premises of why it is important and it is also unclear what the objective of the study really is. For example, the premise for using an RNA-seq approach instead of PCR for the early detection of sex in embryos and foetuses should be developed more, both in the Introduction and also in the Discussion. What are the costs associated with each of these strategies? How long does it take each one from DNA/RNA extraction to final results? I imagine that a PCR can be performed in any regular laboratory as long as a thermo-cycler is available (almost universal), whereas RNA-seq data generation and analysis require a certain NGS expertise. How likely it is that a RNA-seq approach can replace a regular PCR detection, and in what circumstances? It is also unclear why the authors set to develop this RNA-seq approach in pigs. What is the relevance of this animal in comparison with other mammals for example? These premises should be well stated in the Introduction in order to provide a broader context for the work.

The text requires major English editing.

I’m also somewhat concerned with the methods. Can the authors explain why 5 females (6 in the case of ARG at 35 days of gestation) were slaughtered and then only 3 of those were actually used? Also, it is unclear if RNA was extracted from the whole conceptuses or from the respective embryos. Was there any step to remove potential DNA contamination?

L138: it seems that one foetus sample is missing. Was this due to low read quality or any other problem in downstream analyses?

L150-156: these genes seem to have been selected only by looking at the CPM counts but it is unclear if there are significant differences between males and females at these genes or even if other genes also show significant CPM differences between the sexes, as the authors did not apply a formal statistical test. After reading the discussion, I realised that some of these genes were chosen on purpose due to their essential spermatogenic functions (5 of them) but some statistical analyses should be performed for the other 5 "unknown" genes.

Was the MDS plot on Figure drew using data from the whole transcriptome or only from the 10 selected genes?

L244: this study does not demonstrate that the LOC* genes have contributed to sex determination of pig conceptuses, it only shows that these genes are Y-linked, but I guess that was already evident in the genome annotation used by the authors.

Author Response

Date: November 20, 2019

Dear Assistant Editor Ms. Marta Yu and Reviewers,

We appreciated all the comments and suggestions, which allowed us to improve our manuscript E-642615, entitled “Sex determination using RNA-sequencing analyses in early prenatal pig development”. Each comment has been addressed and corresponding changes have been made directly in the manuscript.  All changes in the manuscript were highlighted using track changes and the answers to the reviewers follow below.

Sincerely yours,

Simone E. F. Guimarães

Animal Science Department

Universidade Federal de Viçosa

36.570-000, Viçosa, Minas Gerais, Brazil

Phone number: +55 31 3612 4671

Comments to the reviewers

We appreciate the reviewers comments and suggestions and tried to address all of them as follows:

Reviewer #1

Comments to the Author

Remark: Guimarães et al. use transcriptome data to perform sex determination in the prenatal phase of pigs before conceptuses can be phenotypically differentiated for sex. The study is interesting and can have applications in animal breeding programs. However, it lacks on premises of why it is important and it is also unclear what the objective of the study really is. For example, the premise for using an RNA-seq approach instead of PCR for the early detection of sex in embryos and foetuses should be developed more, both in the Introduction and also in the Discussion. What are the costs associated with each of these strategies? How long does it take each one from DNA/RNA extraction to final results? I imagine that a PCR can be performed in any regular laboratory as long as a thermo-cycler is available (almost universal), whereas RNA-seq data generation and analysis require a certain NGS expertise. How likely it is that a RNA-seq approach can replace a regular PCR detection, and in what circumstances?

Answer: Dear reviewer, we appreciated your comments and suggestions. In this study, we aimed to demonstrate that the sex of conceptuses can be determined from RNA-seq data performed at early pig development stages using a threshold based on the number of reads of chromosome Y-linked genes, normalized by counts per million (CPM), with no need to apply molecular sexing techniques such as qualitative or quantitative PCR. The sex determination is an important step in experimental analysis with animals, since the sex effect can affect the phenotypes commonly studied in livestock. In this context, we aimed to demonstrate that the RNA-seq technique, once applied to obtain a broader context of the transcripts expression level under experimental conditions (e.g. inclusion of additives on maternal diets), can also be applied to determine the conceptuses’ sex. Therefore, this manuscript does not intend to propose a tool to replace the sexing using PCR, but, once RNA-seq has been chosen as the best tool to perform a transcriptome analysis, it is also valuable for other analyses, such as sexual determination in stages that males and females are not phenotypically divergent, since the data is already available.

Following your orientation and suggestions, the Introduction, Discussion and Conclusion sections were reformulated (see L45-52, L61-66, L67-78, L101-107, L240-246, L314-316).

Remark: It is also unclear why the authors set to develop this RNA-seq approach in pigs. What is the relevance of this animal in comparison with other mammals for example? These premises should be well stated in the Introduction in order to provide a broader context for the work.

Answer: In this manuscript, we aimed to determine pig conceptuses sex in early prenatal development using available RNA-seq data. This approach is applicable to other mammalian species of interest for which RNA-seq data has been previously obtained to answer another experimental question and the sex could affect the results, but can not be phenotypically determined. The pig conceptuses RNA-seq data were already available in the present study, since research in our group was interested in the effects of maternal nutrition on fetal programming in pigs. Pigs have been used as an important model in several studies, which may be due to some main factors, such as striking similarities between pigs and humans regarding the anatomy, physiology, metabolism and nutrition [1–3]. Especially in maternal nutrition studies, this similarity is highlighted, since the intrauterine growth restriction (IUGR) is related to the availability of nutrients in maternal diet [4] and affects the expression of key proteins that regulate growth and development of the small intestine, liver and muscles, the major organs involved in digestion, absorption and metabolism of nutrients [3,5], impacting all conceptuses development. In this context, Wu’s group has been performing several studies regarding maternal nutrition of functional amino acids in pigs in order to minimize the IUGR occurrence [6–9]. In addition, the pig is a prolific specie with a relatively short developmental period, besides being a cheaper and easier mammalian model organism to maintain [1,2]. Therefore, the alternative approach proposed in the present study can also be applied during sexing of other mammalian species in which RNA-seq data is already available, with no need for extra sexing technique.  

Following your orientation, we reformulated the Introduction section addressing this issue (see L36-44 and L101-107).

Remark: The text requires major English editing.

Answer: A full English review has been performed and the proofreading certificate is attached.

Remark: I’m also somewhat concerned with the methods. Can the authors explain why 5 females (6 in the case of ARG at 35 days of gestation) were slaughtered and then only 3 of those were actually used?

Answer: We understand your concern. This study is part of a large experiment in which 5 gilts were initially used in the original design described in Costa et al. [10]. However, RNA-seq analysis was performed with 3 gilts in order to reduce costs of molecular analysis, since at least three biological replicates are recommended for describing results from RNA-seq experiments [11,12]. We have clarified this issue in lines 128-132.

Remark:  Also, it is unclear if RNA was extracted from the whole conceptuses or from the respective embryos.

Answer: The conceptuses (embryos from gilts slaughtered at 25 days and fetuses from gilts slaughtered at 35 days) were collected, individually identified and separately macerated in liquid nitrogen. When we mention “whole conceptuses” in the manuscript, we meant that each conceptus was entirely macerated, i.e., we have not chosen a specific tissue, since it was not possible to separate tissues, especially at 25 days after insemination. In order to make it clear, we added a sentence in the manuscript (please see L126-128).

Remark: Was there any step to remove potential DNA contamination?

 Answer: Thank you for your remark. Briefly, for the RNA extraction, we started the protocol using the Trizol reagent, carefully removing the clear upper aqueous containing just the RNA. Then, we followed the protocol using a silica column from RNeasy mini kit protocol (Qiagen), which usually removes some remaining genomic DNA after the buffer washes. Subsequently, we ran a 1% agarose gel and a Bionalyzer chip and we did not observe DNA contamination. We also quantified the total RNA with DNA Qubit kits and no DNA was detected. Finally, our libraries were prepared using mRNA poly-A selection and a stranded protocol with ActD, which enrich the mRNA molecules and avoid DNA amplification. In order to make it clear, we reformulated and added two sentences in the manuscript (please see L145).

Remark: L138: it seems that one foetus sample is missing. Was this due to low read quality or any other problem in downstream analyses?

Answer: Thank you for your remark. The lack of a fetal sample is due to the low RNA Integrity Number (RIN) of this sample (RIN < 8). RIN is a metric developed by Agilent to reduce subjective RNA integrity analyzes based on visual gel analysis, since the quality of the extracted RNA molecules can impact the results of the RNA-seq experiment. This metric is based on the size distribution of RNA molecules in which 1 represent the most degraded and 10 the most intact. In this sense, once several studies have been used RIN greater than 8 as a desirable number of high RNA quality samples, we also have used this threshold in our study and, therefore, removed one sample. We added one sentence about the RIN value of the excluded sample on L155-156 to make it clearer.

Remark: L150-156: these genes seem to have been selected only by looking at the CPM counts but it is unclear if there are significant differences between males and females at these genes or even if other genes also show significant CPM differences between the sexes, as the authors did not apply a formal statistical test. After reading the discussion, I realised that some of these genes were chosen on purpose due to their essential spermatogenic functions (5 of them) but some statistical analyses should be performed for the other 5 "unknown" genes.

Answer: Thank you for your concern. The sex determination of the conceptuses was performed from the read counts normalized to the libraries’ size (Counts Per Million - CPM) of genes mapped on Sus scrofa Y chromosome that were discrepant between samples (high values of reads CPM ​​in one group and almost zero in another, as shown in Tables 1 and 2). The reads CPM were used as count measures, not directly as level of gene expression. Thus, our main aim was to demonstrate the possibility of determining the conceptuses’ sex by a threshold based on discrepant reads counts of Y chromosome-linked genes. Differential expression analysis requires the comparison of gene expression values among samples [11], but for the sex determination this was not the objective. For differential expression purpose, different approaches are used to normalize read count differences for library sizes, as the TMM (Gene-wise trimmed median of means) which is a factor of the libraries normalization, since reads counts  are affected by factors such as transcript length, total number of reads, and sequencing biases [11]. However, as our aim was to perform conceptuses sex differentiation based on the criteria of identification of Y chromosome-linked genes that showed discrepant reads CPM among samples, a differential gene expression analysis, and therefore, a statistical test between male and female groups is not required. We were successful to show this possibility since we found 10 Y chromosome-linked genes that were sufficient to identify and classify the swine conceptuses in males and females in early stages of prenatal development. From these 10 Y chromosome-linked genes, five had a known function associated with spermatogenesis, while the other five genes (LOC*) are not yet fully characterized in pigs. Nevertheless, these 5 LOC* genes were important in our analyses, since they match with the previously criteria (Tables 1 and 2). In order to clarify our objectives and findings, we rewrote the sentences in L101-107, L240-243, L287-290 and L313-315.

Remark: Was the MDS plot on Figure drew using data from the whole transcriptome or only from the 10 selected genes?

Answer: After classification of conceptuses samples as male or females based on discrepant reads of the ten Y-linked genes, the files containing reads counts for all genes mapped against Sus scrofa genome in all chromosomes of our samples (whole transcriptome) obtained by HTseq-count were renamed with sample identification based on sex and experimental condition (arginine or control group) and used as input in plotMDS function of edgeR tool of R software. Therefore, the MDS plot was drew using the files that contained the whole transcriptome of conceptuses obtained from HTseq counts. In order to make it clearer to the reader, we rewrote the sentence about MDS plot (please see L184-186).

Remark: This study does not demonstrate that the LOC* genes have contributed to sex determination of pig conceptuses, it only shows that these genes are Y-linked, but I guess that was already evident in the genome annotation used by the authors.

Answer: Thank you for your comment. The LOC* genes have contributed to sex determination of pig conceptuses since they were located on the Y chromosome and presented discrepant reads CPM among samples (Tables 1 and 2), that were the two criteria used to create a threshold to differentiate males from females in our study. However, the role of these genes during sex determination was not addressed in the present study because these genes have not yet been fully characterized in Sus scrofa. We added a sentence in the Discussion section taking into account your comment (please see L309).

References

Verma, N.; Rettenmeier, A.W.; Schmitz-Spanke, S. Recent advances in the use of Sus scrofa (pig) as a model system for proteomic studies. Proteomics 2011, 11, 776–793. Papatheodorou, I.; Fonseca, N.A.; Keays, M.; Tang, Y.A.; Barrera, E.; Bazant, W.; Burke, M.; Füllgrabe, A.; Fuentes, A.M.P.; George, N.; et al. Expression Atlas: Gene and protein expression across multiple studies and organisms. Nucleic Acids Res. 2018, 46, D246–D251. A, M.E.S.; A, T.S.; A, D.S.G.; A, H.B. Advances in Swine Biomedical Research; Tumbleson, M.E., Schoock, L.B., Eds.; Springer New York, 1996; ISBN ISBN 978-1-4615-5885-9. Costa, K.A.; Marques, D.B.D.; Campos, C.F. De; Saraiva, A.; Guimaraes, J.D.; Guimarães, S.E.F. Nutrition in fl uence on sow reproductive performance and conceptuses development and survival : A review about L -arginine supplementation. Livest. Sci. 2019, 228, 97–103. Godfrey, K.M.; Barker, D.J.P. Fetal programming and adult health. Public Health Nutr. 2001, 4, 611–624. Wu, G.; Bazer, F.W.; Wallace, J.M.; Spencer, T.E. Board-invited review: Intrauterine growth retardation: Implications for the animal sciences. J. Anim. Sci. 2006, 84, 2316–2337. Wu, G.; Bazer, F.W.; Datta, S.; Gao, H.; Johnson, G.A.; Lassala, A.; Li, P.; Satterfield, M.C.; Spencer, T.E. Intrauterine growth retardation in livestock : Implications , mechanisms and solutions. Arch. Tierzucht, Dummerstrof 2008, 51, 4–10. Wu, G.; Bazer, F.W.; Satterfield, M.C.; Li, X.; Wang, X.; Johnson, G.A.; Burghardt, R.C.; Dai, Z.; Wang, J.; Wu, Z. Impacts of arginine nutrition on embryonic and fetal development in mammals. Amino Acids 2013, 45, 241–256. Wu, G.; Bazer, F.W.; Johnson, G.A.; Hou, Y. Board-invited review: Arginine nutrition and metabolism in growing, gestating, and lactating swine. J. Anim. Sci. 2018, 96, 5035–5051. Costa, K.A.; Saraiva, A.; Guimaraes, J.D.; Marques, D.B.D.; Machado-neves, M.; Reis, L.M.B.; Alberto, F.; Veroneze, R.; Oliveira, L.F. de; Garcia, I.S.; et al. Dietary L-arginine supplementation during early gestation of gilts affects conceptuses development. Theriogenology 2019, 140, 62–71. Conesa, A.; Madrigal, P.; Tarazona, S.; Gomez-cabrero, D.; Cervera, A.; Mcpherson, A.; Szcze, W.; Gaffney, D.J.; Elo, L.L.; Zhang, X. A survey of best practices for RNA-seq data analysis. Genome Biol. 2016, 17, 1–19. Nie, H.; Wang, Y.; Su, Y.; Hua, J. Exploration of miRNAs and target genes of cytoplasmic male sterility line in cotton during flower bud development. Funct. Integr. Genomics 2018, 18, 457–476. Pergament, E.; Schulman, J.D.; Copeland, K.; Fine, B.; Black, S.H.; Ginsberg, N.A.; Frederiksen, M.C.; Carpenter, R.J. The risk and efficacy of chorionic villus sampling in multiple gestations. Prenat. Diagn. 1992, 12, 377–384. Alfirevic, Z.; P, von D. Instruments for chorionic villus sampling for prenatal diagnosis. Cochrane Database Syst. Rev. 2013, 2013.

Reviewer 2 Report

Please check the attachments

Author Response

Date: November 20, 2019

Dear Assistant Editor Ms. Marta Yu and Reviewers,

We appreciated all the comments and suggestions, which allowed us to improve our manuscript E-642615, entitled “Sex determination using RNA-sequencing analyses in early prenatal pig development”. Each comment has been addressed and corresponding changes have been made directly in the manuscript.  All changes in the manuscript were highlighted using track changes and the answers to the reviewers follow below.

Sincerely yours,

Simone E. F. Guimarães

Animal Science Department

Universidade Federal de Viçosa

36.570-000, Viçosa, Minas Gerais, Brazil

Phone number: +55 31 3612 4671

Comments to the reviewers

We appreciate the reviewers comments and suggestions and tried to address all of them as follows:

Reviewer #2

Remark: Grammars should be greatly corrected.

Ex. A wide range of traits are sexually dimorphic, indicating that males and females are marked by… (line 44).

Answer: Dear reviewer, we really appreciate your remark. To create a clear and broad context for the study, we have removed this sentence. Additionally, a full English review of the manuscript has been performed and the proofreading certificate is attached.

Remark: 2.     The authors stated “The Y chromosome harbors genes essential for testis development and function, such as the SRY, the master gene for testis” (line 50-51). Is SRY a gene or a region compromising several genes?

Answer: Thank you for your remark. SRY is a gene. We decided to modify the sentence to make it clear and moved to the Discussion section (please see L248-250).

Remark: 3. 3. RNA-sequencing is a new technique and can be applied to a variety of biological studies. The authors applied the technique to examine some sex-specific gene expression for sex determination in prenatal pig embryo at 25 day and fetuses at age of 35 days. Conceptually, this is not a novel idea, and technically the authors did not show any improvements in RNA-sequencing or new tries in sampling.

Answer: Thank you for your comment. I think we have not been clear when stating our purpose with this study in the Introduction section of the manuscript. Our main aim was not to perform a RNA-seq experiment exclusively to examine the expression of some sex-specific genes for sex determination of pig conceptuses at 25 and 35 days-old. Actually, we aimed to demonstrate that the RNA-seq technique, once applied to obtain a broader context of the transcripts expression level under experimental conditions (e.g. inclusion of additives in maternal diets), can also be applied in the determination of conceptuses sex with no need to apply other molecular sexing techniques, as qualitative or quantitative PCR. The sex determination is an important step in experimental analysis with animals, since the sex effect can affect the phenotypes commonly studied in livestock. However, when the study is performed at early developmental stages, this identification becomes more challenging, since male and female conceptuses are not phenotypically divergent. Nevertheless, it is possible to determine the conceptuses’ sex at these gestational stages using the RNA-seq data previously obtained to answer another experimental question. With no knowledge of this possibility, the sex effect would be probably ignored, biasing the transcriptome results. Therefore, the possibility of sex identification from a previously designed RNA-seq study emerges as the novelty of the study, since our knowledge, this is the first study addressing this topic in swine. We have improved our Introduction, Discussion and Conclusion sections in order to clarify our purposes and findings (see L46-52, L61-66, L67-78, L101-107, L240-246 and L313-315).

Remark: Gilts were sacrificed to collect embryos for studies; does it means the application fit to human? Why not try chorionic Villus Sampling of the embryos and save the gilts? since RNA-sequencing is a very sensitive method.

Answer:  We appreciate your concern about the slaughter of gilts. However, chorionic villus sampling tends to increase the risk of miscarriage in several pregnancies as more sampling is required [1,2]. In this context, since the hyperprolific nature of gilts is an important factor in animal science studies to increase the statistical power, given the use of several conceptuses by females (n = 3 / females), the chorionic villus sampling may increase fetal losses. In addition, the design of this experiment was part of a large experiment to study the effects of maternal supplementation on fetal programming and phenotypic data of commercial gilts such as slaughter weight, uterine weight, left uterine horn length, total ovaries weight, embryos number, and viable embryos number, had to be collected from slaughtered gilts, as described in Costa et al. [3]. Therefore, although the conceptuses collection and slaughter of gilts are invasive techniques, they were performed in order to collect gilts phenotypic and biochemical traits and their biological replicates (conceptuses), optimizing the use of these animals. We also emphasize that all experimental protocols used in this study have followed ethical principles in animal research (CONCEA, 2016) and were previously approved by the Ethical Committee on Animal Use of the Universidade Federal de Viçosa (UFV), MG, Brazil [protocol # 06/2017].

Remark: The authors need to highlight new findings and/or methodologies in the study.

Answer: We really appreciate your concern. We improved the Introduction, Discussion and Conclusion sections in order to highlight our findings and methodologies (Please see L55-57, L71-74, L101-107, L240-246, L287-290 and L313-315).

References

Pergament, E.; Schulman, J.D.; Copeland, K.; Fine, B.; Black, S.H.; Ginsberg, N.A.; Frederiksen, M.C.; Carpenter, R.J. The risk and efficacy of chorionic villus sampling in multiple gestations. Prenat. Diagn. 199212, 377–384. Alfirevic, Z.; P, von D. Instruments for chorionic villus sampling for prenatal diagnosis. Cochrane Database Syst. Rev. 20132013. Costa, K.A.; Marques, D.B.D.; Campos, C.F. De; Saraiva, A.; Guimaraes, J.D.; Guimarães, S.E.F. Nutrition in fl uence on sow reproductive performance and conceptuses development and survival : A review about L -arginine supplementation. Livest. Sci. 2019, 228, 97–103.

Round 2

Reviewer 1 Report

The authors addressed all my concerns and I have no more comments on this manuscript.

Reviewer 2 Report

Most of the comments were answered and text were improved.